# The SNP-Based Profiling of Montecristo Feral Goat Populations Reveals a History of Isolation, Bottlenecks, and the Effects of Management

**DOI:** 10.3390/genes13020213

**Published:** 2022-01-24

**Authors:** Elisa Somenzi, Gabriele Senczuk, Roberta Ciampolini, Matteo Cortellari, Elia Vajana, Gwenola Tosser-Klopp, Fabio Pilla, Paolo Ajmone-Marsan, Paola Crepaldi, Licia Colli

**Affiliations:** 1DIANA Dipartimento di Scienze Animali, della Nutrizione e degli Alimenti, Università Cattolica del S. Cuore, 29122 Piacenza, Italy; elia.vajana@epfl.ch (E.V.); paolo.ajmone@unicatt.it (P.A.-M.); licia.colli@unicatt.it (L.C.); 2Dipartimento di Agricoltura, Ambiente e Alimenti, University of Molise, 86100 Campobasso, Italy; g.senczuk@unimol.it (G.S.); pilla@unimol.it (F.P.); 3Dipartimento di Scienze Veterinarie, University of Pisa, 56124 Pisa, Italy; roberta.ciampolini@unipi.it; 4Dipartimento di Scienze Agrarie ed Ambientali—Produzione, Territorio, Agroenergia, University of Milano, 20133 Milan, Italy; matteo.cortellari@unimi.it (M.C.); paola.crepaldi@unimi.it (P.C.); 5Laboratory of Geographic Information Systems (LASIG), School of Architecture, Civil and Environmental Engineering (ENAC), École Polytechnique Fédérale de Lausanne (EPFL), 1015 Lausanne, Switzerland; 6GenPhySE, Université de Toulouse, INRAE, ENVT, F-31326 Castanet-Tolosan, France; gwenola.tosser@inrae.fr; 7Centro di Ricerca Nutrigenomica e Proteomica-PRONUTRIGEN, Università Cattolica del Sacro Cuore, 29122 Piacenza, Italy; 8Centro di Ricerca sulla Biodiversità e sul DNA Antico-BioDNA, Università Cattolica del Sacro Cuore, 29122 Piacenza, Italy

**Keywords:** Montecristo feral goats, SNP, nuclear genome, demographic history, Mediterranean Sea, management

## Abstract

The Montecristo wild goat is an endangered feral population that has been on the homonymous island in the Tuscan Archipelago since ancient times. The origins of Montecristo goats are still debated, with authors dating their introduction either back to Neolithic times or between the 6th and 13th century of the Common Era. To investigate the evolutionary history and relationships of this population we assembled a 50K SNP dataset including 55 Mediterranean breeds and two nuclei of Montecristo goats sampled on the island and from an ex situ conservation project. Diversity levels, gene flow, population structure, and genetic relationships were assessed through multiple approaches. The insular population scored the lowest values of both observed and expected heterozygosity, highlighting reduced genetic variation, while the ex situ nucleus highlighted a less severe reduction. Multivariate statistics, network, and population structure analyses clearly separated the insular nucleus from all other breeds, including the population of Montecristo goats from the mainland. Moreover, admixture and gene flow analyses pinpointed possible genetic inputs received by the two Montecristo goat nuclei from different sources, while Runs of Homozygosity (ROHs) indicated an ancient bottleneck/founder effect in the insular population and recent extensive inbreeding in the ex situ one. Overall, our results suggest that Montecristo goats experienced several demographic fluctuations combined with admixture events over time and highlighted a noticeable differentiation between the two *nuclei.*

## 1. Introduction

For centuries, Mediterranean islands have hosted several feral goat populations, strongly adapted to survive in arid environments and feed scarcity [1]. Among these is the Montecristo feral goat, a free-ranging population inhabiting the homonymous Italian island since ancient times. The Montecristo Island is a relatively small island (ca. 1000 hectares) in the Mediterranean basin. Located in the Thyrrenian sea 60 kms from the coasts of Tuscany, this island has been a Nature Reserve since 1971 [2,3], and hosts several endangered and endemic species. The Montecristo feral goats are characterised by phenotypic traits shared both with the domestic goat *Capra hircus* (i.e., the small size and the wide colour variability) and with the semi-wild goat populations of the Mediterranean basin (i.e., scimitar shape horns, present in both sexes but more prominent in males) [4,5]. Due to the occurrence of mixed phenotypic traits, the taxonomic status of Montecristo feral goats has long been debated, with some authors referring to this population as *Capra aegagrus,* to point to a closer relationship with the Mediterranean feral goats [6].

Despite several hypotheses having been proposed, the origins of the Montecristo goats are unknown; indeed, any precise inference has been hampered so far by the lack of clear archaeological data owing to the island soil that prevents the formation of fossils [7]. Some authors [8] have suggested that Montecristo goats were introduced onto the island during Neolithic times, while others [9,10,11] date the first occurrence much more recently between the 6th and 13th centuries, when the goats were exploited as a food resource by the monk community settled on the island. In the mid-19th century the presence of feral goats on the island was reported by Alexandre Dumas in his famous novel “The Count of Montecristo”: “*At every step that Edmond took he disturbed the lizards glittering with the hues of the emerald; afar off he saw the wild goats bounding from crag to crag*”, and was further documented by the naturalists exploring the island in the same period [6]. From the late-19th century until the mid-20th Montecristo Island was exploited as a Savoia royal family’s game reserve. In the same period, the introduction of goats from the mainland for restocking purposes was reported [12]. Between the 1950s and the 1960s the island was owned by a private company, which established a game reserve [6]. Due to excessive hunting the original nucleus of goats was possibly reduced to less than 10 individuals [10] and subsequently restocked with goats from the mainland [6]. When Montecristo Island was declared a Nature Reserve in 1971 to preserve the autochthonous species and the local goats [2], hunting was finally forbidden, leading to a steady increase in the number of goats, which soon became a threat to the island ecosystem. Since then, several selective culling campaigns have been performed to reduce population size [12].

Currently, a small population of goats are farmed in Tuscany in the province of Grosseto, which are described as the descendants of animals allegedly moved from Montecristo Island to the Italian mainland in the last decades of the 20th century for conservation and research purposes. The information regarding the origin, recent history, and management of these nuclei is scanty, with their establishment being dated back either between the end of the 1970s and 1980s, or even before the Nature Reserve was set up in 1971 [13]. These small nuclei of animals, i.e., each one including one male and four females, were hosted at different breeding facilities on the mainland (Ciampolini R., pers. comm.) and likely underwent strong demographic fluctuations in the following decades.

Starting from the last decade of the 20th century, a few studies have been carried out to characterize Montecristo goats by means of molecular markers. In 1990, Randi and colleagues [13] analysed allozyme loci of 20 samples from the Montecristo goat population, and underlined the occurrence of several introductions of domestic goats from mainland that contributed to the original gene pool. More recently, Doro and colleagues [4] analysed the complete mitochondrial DNA sequence of a single male specimen of Montecristo feral goat, highlighting a similarity with Western European domestic lineages. Analyses of Montecristo goats microsatellites and mtDNA were part of a LIFE project focussed on Montecristo island [14], which highlighted the absence of bottleneck events in the insular population as well as the presence of two unique mitochondrial haplotypes. Furthermore, a clear differentiation between insular and ex situ stocks was assessed [12]. The animals sampled on the mainland, in fact, were not assigned to the gene pool typical of insular Montecristo goats, since they displayed mitochondrial haplotypes and microsatellite alleles not found in the insular population [14].

Several authors have discussed the origin and the conservation status of the Montecristo goats during the last decades [6,15]. However, the combined lack of pre-1980s demographical information and extensive genomic sampling has not allowed light to be shed on the genetic heritage of this feral stock, so far.

To tackle this gap, the present work assembled and analysed a 50K SNP dataset including 55 domestic goat breeds and 50 Montecristo individuals, from both the island and the mainland, to investigate the demographical history of the Montecristo population in the context of the Mediterranean basin. To our knowledge, this is the most comprehensive study on the Montecristo feral goat molecular diversity to date, which can represent a first step towards marker-assisted conservation of this peculiar goat population.

## 2. Materials and Methods

### 2.1. Dataset Construction and Filtering

The Montecristo goat samples included 32 individuals (MNT_I) from the free-ranging population inhabiting Montecristo Island sampled between 1995 and 2012, and 18 goats (MNT_M) reared ex situ at a farm in Seggiano on the Mount Amiata (Grosseto province, Tuscany, Italy). The latter population was sampled in 2016 and comprises the descendants of two of the four starting nuclei of individuals allegedly moved from the island of Montecristo to the Italian mainland between the end of the 1970s and the beginning of the 1980s.

The animals were genotyped with the Illumina GoatSNP50 BeadChip [16] as described in Cortellari et al. [17]. Genotypes were merged with publicly available data representing 55 goat breeds from the Mediterranean basin and south-western Asia (Figure 1; Appendix A) [18,19,20].

Data from different sources were re-mapped on the goat reference genome ARS1, merged, and quality-controlled using PLINK 1.9 [20]. Individuals and markers exceeding the following thresholds were removed: (i) SNP call rate < 0.98; (ii) individual genotype call rate < 0.96; (iii) minor allele frequency (MAF) < 0.1. SNPs with unknown map position or located on sex chromosomes were removed. Pruning for linkage disequilibrium (LD) was performed using the *‘--indep-pairwise’* function in PLINK [20], where SNPs with r^2^ > 0.2 were removed from sliding windows of 50 SNPs and a step size of five SNPs. Breeds with sample size larger than 30 individuals were thinned to a subset of 30 representative samples using the function *representative.sample* implemented in R package BITE [21].

To mitigate the bias possibly deriving from the reduced number of polymorphic loci in MNT_I population, we assembled a second dataset including only the markers scored as polymorphic in MNT_I (poly-MNT_I dataset). The two datasets were subjected to the same analyses and the results were compared.

### 2.2. Estimation of Genetic Diversity, Population Structure and Migration Events

Observed and Expected heterozygosities (H_O_ and H_E_) and the inbreeding coefficient (F_IS_) values were calculated using the software Arelquin v3.5.2.2, which allows analysing genomic data [22]. To minimize the effects of the different numbers of polymorphic loci on the estimates of diversity, heterozygosity values were corrected over the number of usable SNPs with the formula proposed by Colli and colleagues [23].

Principal Component Analysis (PCA) was performed with the ‘*--pca-clusters*’ flag in PLINK v1.9 [20] in both and unsupervised (i.e., standard) and a supervised fashion to account for outlier’s behaviour, using the ‘*--pca-clusters*’ function. In this approach, outlier populations are first identified through unsupervised PCA, then principal components are calculated leaving the outlier populations out in the supervised analysis; finally, the individuals belonging to the outlier populations are assigned coordinates and projected onto the supervised PC axes. Results were visualized in R v3.6.1 [24].

Reynolds’ unweighted distances between breeds were calculated with Arlequin [22] and used to build a Neighbour-net graph with SplitsTree v4.14.6 software [25]. Admixture v 1.3.0 [26] was used to evaluate population structure through a maximum-likelihood-based approach. Analyses were performed for K values ranging from 2 to 45. The best fitting K was identified as the one scoring the lowest cross-validation error value.

The occurrence of migration events was investigated with the software Treemix v. 1.13 [27] by setting windows of 500 consecutive SNPs and testing migration events (*m*) from 0 to 11, with 5 iteration each. An ad hoc statistic implemented in the R package OptM was used to identify the most likely number of migration edges [28]. The robustness of the nodes of Treemix underlying graph was estimated through 100 bootstrap replicates run for the best *m* value. A consensus tree was produced using the *consense.exe* executable in PHYLIP v3.695 [29] and plotted with the BITE function *treemix.bootstrap* [21].

Lastly, we used the LD-based method implemented in the SNeP v1.1 software [30] to evaluate the changes in Effective Population size (*Ne*) during the last 1000 generations. This analysis was performed on the two Montecristo feral goats’ populations MNT_I and MNT_M, and on the three geographically closest breeds, namely, Garfagnana (GRF), Sarda (SAR) and Corse (CRS).

### 2.3. Runs of Homozygosity and Heterozygosity-Rich Regions

Continuous stretches of homozygous sequences, i.e., Runs of homozygosity (ROHs), were scored on the two Montecristo *nuclei* to assess population demographic history. ROHs were computed with PLINK v1.9 [20] as described in [31]. Number of ROHs per animal, average ROH length and ROH distribution across length classes (0–2 Mb, 2–4 Mb, 4–8 Mb, 8–16 Mb, and >16 Mb; see Appendix A) were calculated. Scored ROHs were visualized chromosome-wise in R v3.6.1 [24] and the ROH coverage-derived genomic inbreeding was computed with the dedicated function in R package detectRUNS [32]. The per-proportion of times each SNP falls inside a run in a given population was calculated using the R package detectRUNS [32]. SNPs with a value in the top 0.1% of the percentile distribution were considered as statistically significant. Each significant SNP was annotated with the R package GALLO [33], considering an interval of 1000 bp upstream and downstream the examined marker. Genomic inbreeding derived from ROH coverage was computed for the two Montecristo populations using the dedicated formula implemented in the package detectRUNS [32]. Heterozygosity-rich regions (HHRs) were evaluated with the R package detectRUNS through the “Sliding Windows” method with the following criteria: (i) the window size was set to 10 SNPs; (ii) the window threshold was kept with the default 0.05 value; (iii) the minimum number of SNPs in a HHR was 5; and (iv) the minimum length of a HHR was 500 bps. For identifying shared regions of interest, the same approach used for ROHs was applied. The top 0.1% SNPs in the percentile distribution of the number of times each SNP falls inside a run were considered as to be significant. Significant markers were annotated with the R package GALLO [33].

### 2.4. Approximate Bayesian Computation

The Approximate Bayesian Computation–Random Forest approach (ABC-RF) implemented in the DiyABC—rf1.0 software [34] was used to test alternative scenarios describing the recent demographic history of Montecristo goats. This approach allows to: (i) simulate multiple historical models; (ii) evaluate the parameter estimated during the analysis; and (iii) rank the best fitting model based on the approximate posterior probabilities.

A dedicated dataset was assembled to carry out ABC-RF analysis, including the two Montecristo populations (MNT_M, MNT_I) together with the geographically and genetically closest breeds as identified in previous analyses: Garfagnana (GRF), Sarda (SAR) and Corse (CRS). To reduce computational burden the dataset was further pruned for LD, reducing the number of SNPs to 9757. To estimate the extent of a possible loss of information in the reduced dataset, we computed the Pearson’s correlation coefficient between eigenvector values obtained with the main and the reduced datasets.

To test our hypotheses on recent demographic history, we modelled four different scenarios based on the results of population structure analyses and available historical records on the Montecristo goat population. In Scenario 1 (Appendix A, panel a) we tested the hypothesis of the continental population (ancestors of GRF) diverging as first from the insular ones. The Montecristo goat population was modelled as deriving from an admixture event involving the Sarda and Corsa domestic breeds. Under Scenario 2 (Appendix A, panel b) we assumed a more ancient origin of the Montecristo goat population, which diverged before the split between the Sarda and Corse breeds. Scenario 3 (Appendix A, panel c) was designed to test a different origin of the two Montecristo populations with the insular one originating from the Corse and the mainland one from the Sarda breed. In Scenario 4 (Appendix A, panel d) a possible derivation of the Montecristo population from the Corse breed was tested.

## 3. Results

### 3.1. Dataset Construction and Filtering

After quality control routines, a working dataset including 1251 animals and 43,252 SNPs was retained for subsequent analyses; while the number of markers was further reduced to 33,123 SNPs in the poly-MNT_I dataset containing only the markers that were polymorphic in the MNT_I population. The comparison between the results obtained from the main dataset and the poly-MNT_I datasets showed no substantial differences. Therefore, only the results obtained from the main working dataset were shown here.

### 3.2. Estimation of Genetic Diversity, Population Structure, and Migration Events

The Observed heterozygosity (H_O_) corrected value calculated for the insular Montecristo goats (MNT_I) was the lowest recorded in the dataset (i.e., H_O_ = 0.271), and lower than that of the bezoar (H_O_ = 0.281) (Appendix A). The continental population MNT_M, instead, scored a higher value (H_O_ = 0.324), which is not far from the lower end of the range of H_O_ values scored for the domestic goat breeds (0.348 (VLS) < H_O_ < 0.417 (MLG). The corrected values of Expected Heterozygosity (H_E_) were higher than H_O_ for both Montecristo populations (MNT_M: H_E_ = 0.377 vs. H_O_ = 0.324 and MNT_I: H_E_ = 0.347 vs. H_O_ = 0.281), with the insular population scoring the lowest value in the dataset. For the whole dataset the *p*-values associated to the inbreeding coefficient F_IS_ estimates were not statistically significant, except for the bezoar F_IS_ = 0.233 (significant at *p* < 0.005. Appendix A).

The first and second Principal Components of the unsupervised PCA together accounted for 5.57% of the total variance (Figure 2). On PC1 (3.99% of variance) the breeds were distributed following a seamless north–south geographical pattern, with the northern Italian populations on the left side of the plot and the northern African ones on the right side (Figure 2). The MNT_M population was positioned close to two Spanish breeds, Murciano Granadina (MUG) and Malagueña (MLG).

The second PC (1.58% of variance) highlighted an extreme outlier behaviour of the insular Montecristo goats MNT_I, which lies at the bottom left corner of the plot at a great distance from all other populations (Figure 2).

As detailed previously, the supervised Principal Component Analysis (Figure 3; Appendix A) was then carried out to minimize the impact of outlier populations. In this analysis, SPC1 (4.1% of variance) clearly separated the European breeds from the African and south-western Asian ones. Together, SPC1 and SPC2 (5.5% of variance explained overall) showed a clear clustering of the breeds based on their country of origin and confirmed a north-to-south genetic pattern within Italy, as already highlighted in previous studies [18,19]. The insular Montecristo goat population MNT_I clustered with breeds from central Italy (GRF, FAC, FUL), Sardinia (SAR), and Corse (CRS), while the scatter of points belonging to the ex situ MNT_M individuals was placed close to the origin of the axes and stretched between the SAR and Moroccan populations (Figure 3).

The Neighbour-network based on Reynolds genetic distances (Figure 4) confirmed the geographical structuring of diversity already pointed out by PCA analyses. Both Montecristo goat populations were positioned on long branches, usually interpreted as evidence of prolonged isolation likely combined with genetic drift. Similar to SPCA results, the MNT_I population clustered with the Sarda, Corse, and Garfagnana breeds, while the MNT_M population was with some central Italian (FAC) and Spanish breeds (MLG, MUG, RAS, and BEY).

Among the tested admixture scenarios (K = 2 to K = 45; Figure 5), the best fitting resolution was identified at K = 31 according to the CV error decay (Appendix A).

The model assuming two ancestral populations (K = 2) firstly separated the populations on a geographical basis, with the breeds from European assigned to a cluster different from that of African and south-western Asian ones. Already at this low K, the two Montecristo populations showed different behaviours: MNT_I was clearly assigned to the European cluster, while MNT_M displayed an admixed genomic background including both ancestral components. At K = 3 the insular MNT_I population clearly clustered apart from all other breeds. Conversely MNT_M (i) showed the occurrence of a genomic component shared with the breeds from northern Africa, southern France and Spain at K = 5; (ii) was assigned to a separated cluster at K = 6; and (iii) showed a sub-structure at K = 30 (Figure 5).

Regarding the Treemix analysis, the Evanno statistic calculated over five iterations for *m* from 0 to 10 indicated *m*6 as the most likely number of gene flow events (Appendix A). In the corresponding tree-based graph, most of the nodes were supported by high bootstrap values (Figure 6). Several migration edges connected the northern African breeds with each other and with those from Spain and southern France. MNT_I was positioned on the same branch as CRS, the latter being also connected to the MNT_M basal node by a migration edge.

SNeP analysis pinpointed a marked but gradual reduction in effective population size over time for both MNT_I and MNT_M. The decline in *Ne* was consistent among the two Montecristo populations and less steep compared to the behaviour of the domestic SAR, CRS, and GRF (Appendix A).

### 3.3. Runs of Homozygosity and Heterozygosity-Rich Regions

The average number of ROHs per animal was 239.7 and 66.35 for MNT_I and MNT_M, respectively, and the total number of ROHs identified was 1128 for the MNT_M and 7191 for the MNT_I population. ROHs of the two populations were classified into five length classes (0–2 Mb, 2–4 Mb, 4–8 Mb, 8–16 Mb, and >16 Mb. Appendix A): in the MNT_I population, the highest number of ROHs was scored in the shortest length class (0–2 Mb), with the number of ROHs in the remaining classes gradually decreasing to 71 ROHs in the >16 Mb class. Conversely, the MNT_M population showed a more homogeneous distribution of ROHs across the five length classes, with the highest frequency in the 4–8 Mb class (Appendix A) and with an occurrence of ROHs >16 Mb class almost as frequent as in the other classes. The values of genomic inbreeding derived from the ROH coverage were 0.270 for the MNT_M and 0.312 for the MNT_I.

The scored ROHs were plotted chromosome-wise to obtain further insight on the contrasting behaviour of the two Montecristo goat populations, which showed a clear difference in the genomic distribution of the homozygous stretches (Appendix A). The MNT_I individuals, in fact, showed the occurrence of generally short ROHs uniformly distributed along the chromosomes and interspersed with heterozygous stretches, and without major differences between individuals. The MNT_M animals conversely displayed a highly variable behaviour in terms of ROH occurrence, length, and position. Some individuals showed uninterrupted ROHs spanning over long chromosome tracts, sometimes as long as >80% of the chromosome (e.g., see chr4) and accompanied by very extended and completely heterozygous regions (e.g., chr6 and chr19) (Appendix A). Moreover, the single-chromosome plots also pointed at the occurrence of regions consistently homozygous/heterozygous across individuals. This evidence was further investigated by identifying ROH regions including the top 0.1% of the most represented SNPs, which were found on chromosomes 1, 2, 3, and 11 for MNT_I, and on chromosome 16 for MNT_M (See Appendix A). In MNT_I shared ROH islands harboured the genes *TNP1*, *SMARCAL1*, *MARCHF4*, and *ST6GALNAC5*, while for the MNT_M population the highly shared ROH region on chromosome 16 included the genes 5S_rRNA, *RRP15*, *KCTD3*, *BRINP3*, *GPATCH2*, *ESRRG*, *TGFB2*, and S*PATA17*. Several unannotated sequences were also found in the ROHs of both populations.

For MNT_I common HRRs were found on chromosomes 6 and 13 (Appendix A). The HRR on chromosome 6 included the genes *PPP3CA*, *EMCN*, *CNOT6L*, *SHLD1*, and *GPCPD1*. *HHRs* shared by MNT_M individuals were located on chromosomes 1, 2, 8, 13, 14, 16, 18, 22, and 24 and spanned the genes *OTOL1*, *LYPD6B*, *NFIB*, *BMP1*, *OPTN*, *BEND7*, *MCM10*, *PHYH*, *ASAP1*, *ADCY8*, *TATDN3*, *RPS6KC1*, *ANGEL2*, *NSL1*, *LOXHD1*, *KATNAL2*, *ST8SIA5*, *PIAS2*, *SKOR2*, and *SMAD4*.

### 3.4. Approximate Bayesian Computation

According to the value of the Pearson correlation coefficient calculated between the main vs. reduced dataset PCA loadings (0.971), loss of information due to the reduction in the number of SNPs was excluded. Scenario 2 (divergence of the Montecristo goat population before the split between the Sarda and Corse breed) resulted in the most supported one (64.8% of the votes in model choice prediction). However, the simulations never fit the observed data (Appendix A), which suggests that none of the four models devised could thoroughly account for the complex demographic history of the Montecristo goats.

## 4. Discussion

In this study, we performed the first genome wide assessment of the genetic variation in Montecristo feral goats to shed light on their levels of polymorphism, population structure, and demographic history, and to evaluate their historic relationships with goat breeds across the Mediterranean area.

Previous investigations on the origin of Montecristo feral goats were based either on phenotypic traits, or on a limited number of allozymic/microsatellite loci or on partial fragments of the mtDNA control region [4,14,16]. Here, the availability of new genotype data from two different *nuclei* of Montecristo goats, i.e., the free-ranging insular population and the captive-bred nucleus, also enabled us to compare their genomic make-up and to evaluate the effectiveness of the ex situ conservation project that has been carried out on the Italian mainland since the last decades of the 20th century. Overall, the results consistently highlighted marked molecular differences between the in situ and ex situ nuclei. The in situ population showed the lowest recorded value of observed heterozygosity in the dataset (Appendix A), a strong outlier behaviour in the unsupervised PCA (Figure 2), a clear independent clustering in admixture analysis (Figure 5), and a high value of genomic inbreeding (F_ROH_ = 0.312). These findings are in line with expectations, considering the geographical isolation and the repeated bottleneck events that have characterised the demographic history of the insular population, at least from the 18th century. They also agree with previous studies on insular populations, i.e., goats from the Mediterranean basin [35] and Soay sheep from the Scottish island of Hirta [36,37]. In all cases, insular populations had increased levels of inbreeding and a reduced variability compared to the nearby mainland populations. ROH analysis identified a high frequency of short ROHs that can be traced back to ancient bottlenecks and founder effects (Appendix A). Moreover, in the supervised PCA (Figure 3), Neighbour-network (Figure 4), and Treemix graph (Figure 6), the insular MNT_I population showed a genetic proximity to local breeds from Central Italy and the nearby islands of Corse and Sardinia, as expected based on the geographical location of Montecristo Island in the Tuscan archipelago, and the recorded inputs of domestic stocks during the 20th century.

Taken together, the results obtained from the insular population account for an ancestral genomic background shared with domestic breeds of the Thyrrenian sea area, subsequently moulded by a history of prolonged isolation, with several ancient bottlenecks likely accompanied by gene flow events from domestic stocks mainly from neighbouring areas.

Conversely, the population from the mainland displayed higher observed heterozygosity (MNT_M H_O_ = 0.325) and lower genomic inbreeding (F_ROH_ = 0.270) compared to the insular population (MNT_I H_O_ = 0.272; F_ROH_ = 0.312). The inbreeding level of the mainland population is remarkably higher than in domestic goat breeds from Italy [38] and other areas of the world [39,40], while observed heterozygosity is only slightly lower than those observed in other breeds.

This is the result of the peculiar demographic history of the ex situ population. Four original ex situ nuclei were established, each one including only one male and four females, which inevitably contributed to a reduced amount of starting genetic variation. The nuclei were hosted at different farms and bred separately, without exchange of males and females for several generations (Ciampolini R., per. comm.). This led to a fast increase in inbreeding due to prolonged reproductive isolation, as testified by the extensive occurrence of long Runs of Homozygosity.

A few years ago, the only two remaining nuclei were finally merged into a single population (Ciampolini R., per. comm.). This recent merging is the likely cause of the higher-than-expected H_O_ in the mainland population, as an increase in observed heterozygosity is expected to occur following the removal of a reproductive barrier between two formerly separated populations, particularly when they are highly inbred and heavily affected by genetic drift due to small number of founders and small population size. This is further confirmed by the occurrence of long stretches of completely heterozygous regions (Appendix A).

The peculiar genomic makeup of the ex situ population, characterised by a mosaic of alternated long stretches of completely homozygous/heterozygous regions in most of the individuals (Appendix A), likely affected the behaviour of MNT_M in several analyses, making the identification of the actual relationship with the insular population and the other breeds difficult. The PCA/sPCA, population structure and neighbour-network results (Figure 2, Figure 4 and Figure 5), in fact, pointed to an affinity between the ex situ nucleus and the Italian Sarda, the Spanish and northern African breeds. This evidence is not consistent with the behaviour of MNT_I in the same analyses, nor with the known history of MNT_M after the establishment of the ex situ nuclei, even if the possible occurrence of unrecorded crossbreeding events with other domestic goats (e.g., Sarda or central Italian breeds) cannot be ruled out.

The more supported scenario obtained from the ABC analyses accounted for a first separation of the central Italian Garfagnana (GRF), followed by that of the Montecristo populations, and later by that of the Sarda (SAR) and Corse (CRS) breeds. This scenario would suggest a shared ancestry between the two MNT populations and their common, but more ancient, origin with the Sardo–Corse domestic stocks. However, these results are to be taken with caution, as simulated data did not fit the observed ones (Appendix A), possibly because of the MNT_M peculiar genetic makeup. Overall, current results are not conclusive on the relationships between the two Montecristo goat populations and further analyses are needed, possibly based on whole genome sequence and the analysis of haplotype blocks.

Conversely, some interesting genes were mapped in the most shared ROH/HRR regions. In MNT_I, the most shared ROHs harboured, among others, the genes TNP1 and SMARCAL1, respectively associated to spermiogenesis [41] and genome integrity [42] (Appendix A). The highly shared HRR on chr6, instead, harboured the gene PPP3CA, previously associated to fecundity traits and litter size in small ruminants [43,44], while the gene GPCPD1 found in chr13 HRR was identified as playing a potential role in the metabolism of lipids and the lipoprotein pathway in sheep [45]. These results may suggest that both the fixation of specific variants in genes related to reproduction and integrity of the genome on the one hand, and the maintenance of consistent heterozygosity of genes involved in fecundity and energy metabolism on the other hand, may have played a role in the adaptation to the harsh environment of the island despite recurrent demographic fluctuations.

Conversely, molecular evidence points to a fixation of variants in genes involved in disease resistance, which may be alarming, and to the retention of heterozygosity in genes related to the development of internal organs and fertility in the case of the Montecristo goat population from the mainland.

## 5. Conclusions

In this study, we present the first comprehensive analysis of the demographic history and genome-wide molecular variation in Montecristo feral goats. According to our results, the insular population faced several demographic fluctuations over the centuries, which partially eroded the original genomic make-up in combination with gene flow events from other goat breeds. This population shares its ancestry with breeds from the surrounding areas of Sardinia, Corse, and Tuscany and does not carry signatures of recent inbreeding. Conversely, the molecular diversity of the mainland Montecristo goat population seems to have been severely impacted by the dynamics of the ex situ breeding that confounds the reconstruction of past population history. Overall, our findings represent a starting point for the implementation of marker-assisted monitoring and conservation plans to preserve the genomic heritage of the feral goats from Montecristo Island.

## Figures and Tables

**Figure 1 genes-13-00213-f001:**
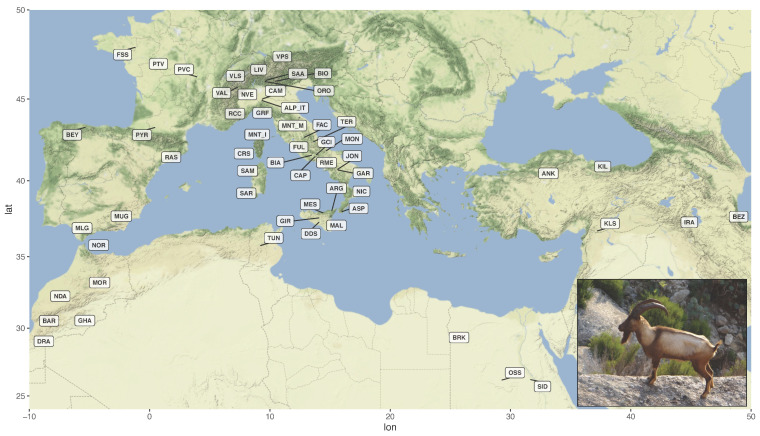
Geographical distribution of the 57 breeds included in the dataset. The labels indicate the centroids of sampling locations. The inset shows a male Montecristo goat with light-coloured phenotype (Photo from https://www.ruminantia.it/vi-raccontiamo-le-razze-la-capra-di-montecristo/ accessed on 11 November 2021). For the correspondence between labels and breed names refer to Appendix A.

**Figure 2 genes-13-00213-f002:**
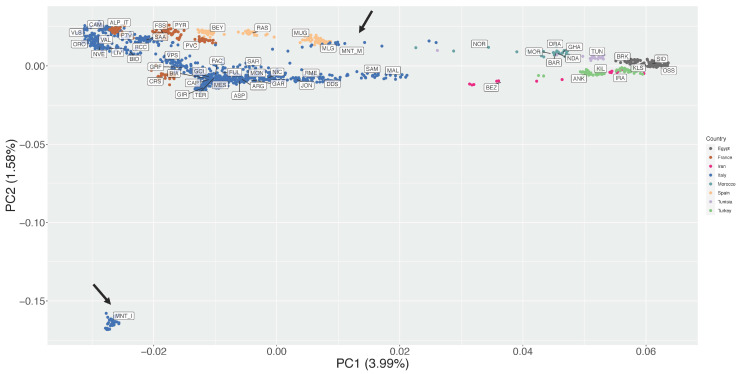
Unsupervised Principal Components Analysis (PC1 vs. PC2). The percentages of variance explained by each component are given into brackets. Arrows indicate position of Montecristo populations in the figure.

**Figure 3 genes-13-00213-f003:**
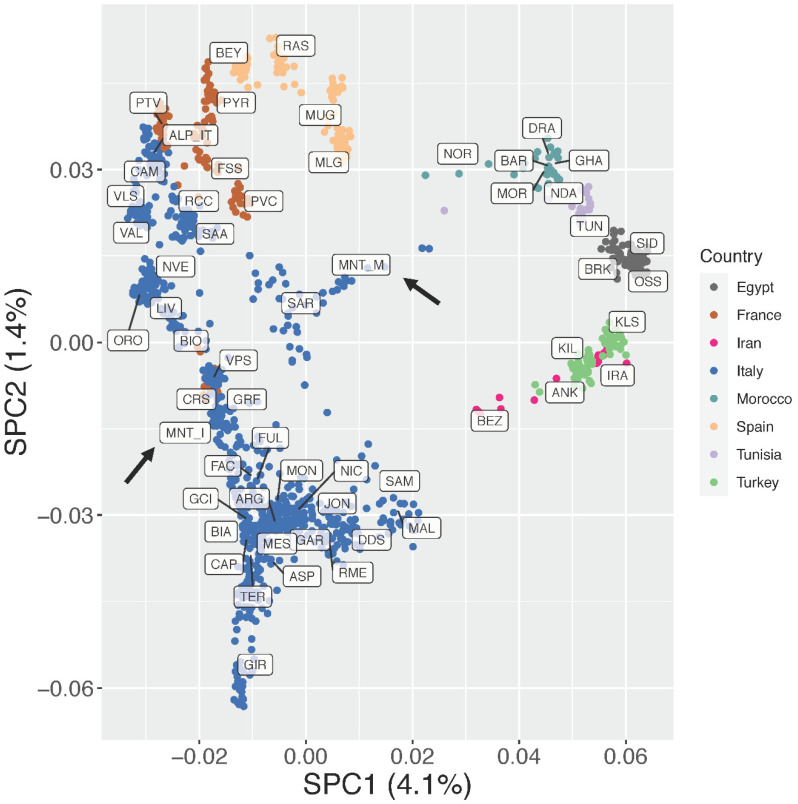
Supervised Principal Component Analysis (SPC1 vs. SPC2). The percentages of variance explained by each component are given into brackets.

**Figure 4 genes-13-00213-f004:**
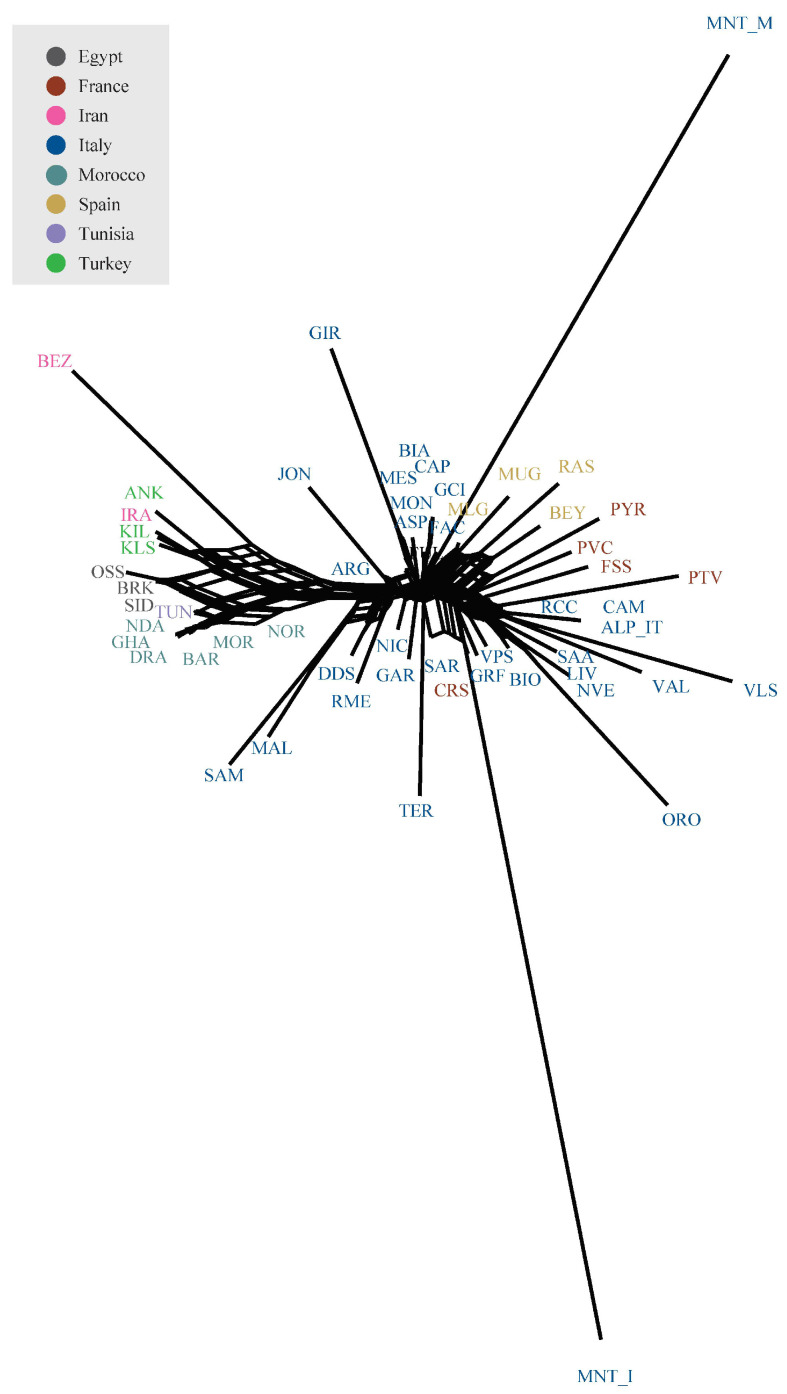
Neighbour-network reconstruction based on Reynolds’ genetic distances between breeds.

**Figure 5 genes-13-00213-f005:**
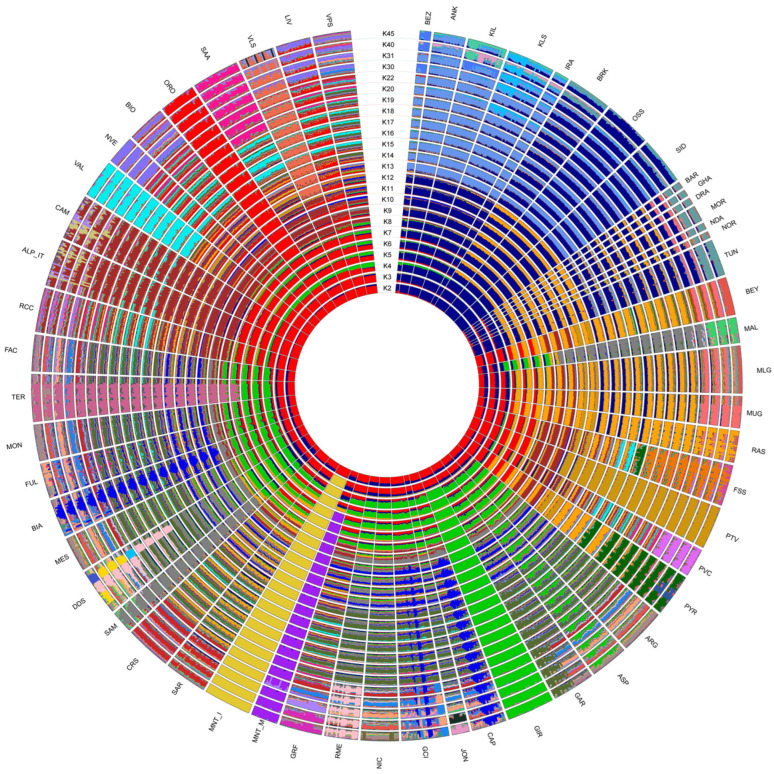
ADMIXTURE software analysis with putative ancestral population (K) computed from 2 to 45. The reconstruction at K = 31 had the smallest cross-validation error.

**Figure 6 genes-13-00213-f006:**
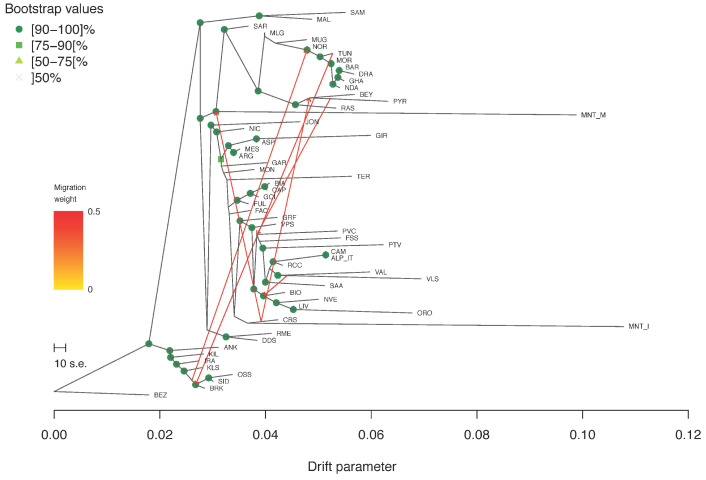
Treemix graph corresponding to *m* = 6. Robustness of the nodes was computed over 100 bootstrap replicates.

## Data Availability

The genotyping data used in this study are deposited and publicly available on Mendeley Data (DOI: 10.17632/hnd59×6gmg.1; URL: https://data.mendeley.com/datasets/hnd59×6gmg/1 accessed on 11 December 2021) and on DRYAD (URL: https://datadryad.org/stash/dataset/doi:10.5061/dryad.v8g21pt accessed on 11 December 2021).

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
