# Peer review of "The SNP-Based Profiling of Montecristo Feral Goat Populations Reveals a History of Isolation, Bottlenecks, and the Effects of Management"

_genes, 2022, doi:10.3390/genes13020213_

Round 1

Reviewer 1 Report

The aim of the research was to conduct a genetic analysis of Montecristo goat population and searching for the origins of Montecristo goats.

The research was carried out on samples taken from Montecristo goats inhabiting the island in the Tuscan Archipelago and goats of this breed bred on the mainland, as well as from other breeds of goats inhabiting the mainland from the Mediterranean region. The study investigated SNP polymorphism using 50K SNP microarrays. The analysed goat population is an interesting subject of research. Presumably, the Montecristo goats have been living on the island for many years, without the possibility of crossing with other species of goats. The authors of the study hoped to find an answer to the question: for how many years have Montecristo goats inhabited the island and whether it is possible to identify ancestors of this breed in the current population of goats on land.

The job is very interesting. It makes a major contribution to the field of genetic biodiversity and phylogenetic networks studies of animal populations that have been geographically isolated over the years.

I only have one remark to the study. In the chapter "Discussion" lacks of information about studies  in which similar issues were undertaken but on other farm animals, for example sheep from the Soay island.

I would also like to suggest a colour change in the markings for sheep from Morocco. The yellow colour in Fig 3 is not visible.

Reviewer 2 Report

Paper is written in structure and quality expected from such type of works.  It has all formal parts requested from the guide for authors and uses standard annotation of references in the Introduction or subsequent Material and Methods section. 

With this I would stress to take care when citing software (pls.see L.147) reference [22] or L.122  - Cortellari et al. (2021). But those are of lower importance and only minor.

Generally, material and methods description is standard as expected in such type of submission. I was quite surprised because not seeing use of Arlequin in scientific paper for last 10 years. All the statistics can be directly made with use of scripts compiled in R. Previous versions were limited on number of SNP used in analyses, therefore short paragraph for clarification of use Arlequin would be of benefit for scientific audience. 

Consequently, I am missing why Reynolds distance were preferred over standard Nei distances which are more common in the similar studies - is it because they were calculated in Arlequin? Therefore short paragraph for clarification would be of benefit. 

Finally, I am missing information why GALLO was preferred by Authors over other line ensembldb.

Overall, the submission is original, written in high quality and presentation is very good, scientifically sound with possibly high interest for the readers.
